# Habitat loss weakens the positive relationship between grassland plant richness and above-ground biomass

Yongzhi Yan[1], Scott Jarvie[2], Qing Zhang[1,3,4]*

[1]Ministry of Education Key Laboratory of Ecology and Resource Use of the Mongolian Plateau, School of Ecology and Environment, Inner Mongolia University, Hohhot, China; [2]Otago Regional Council, Dunedin, New Zealand; [3]Collaborative Innovation Center for Grassland Ecological Security (Jointly Supported by the Ministry of Education of China and Inner Mongolia Autonomous Region), Hohhot, China; [4]Autonomous Region Collaborative Innovation Center for Integrated Management of Water Resources and Water Environment in the Inner Mongolia Reaches of the Yellow River, Hohhot, China

*For correspondence:
qzhang82@163.com

**Abstract** Habitat loss and fragmentation per se have been shown to be a major threat to global biodiversity and ecosystem function. However, little is known about how habitat loss and fragmentation per se alters the relationship between biodiversity and ecosystem function (BEF relationship) in the natural landscape context. Based on 130 landscapes identified by a stratified random sampling in the agro-pastoral ecotone of northern China, we investigated the effects of landscape context (habitat loss and fragmentation per se) on plant richness, above-ground biomass, and the relationship between them in grassland communities using a structural equation model. We found that habitat loss directly decreased plant richness and hence decreased above-ground biomass, while fragmentation per se directly increased plant richness and hence increased above-ground biomass. Fragmentation per se also directly decreased soil water content and hence decreased above-ground biomass. Meanwhile, habitat loss decreased the magnitude of the positive relationship between plant richness and above-ground biomass by reducing the percentage of grassland specialists in the community, while fragmentation per se had no significant modulating effect on this relationship. These results demonstrate that habitat loss and fragmentation per se have inconsistent effects on BEF, with the BEF relationship being modulated by landscape context. Our findings emphasise that habitat loss rather than fragmentation per se can weaken the positive BEF relationship by decreasing the degree of habitat specialisation of the community.

## eLife assessment

This **important** study advances our understanding of how landscape context affects the relationship between grassland plant diversity and biomass. This study used very well-designed approaches to analyze complex ecological relationships in real-world landscapes and thus provides **compelling** evidence to support its findings. The work will be of interest to landscape ecologists and community ecologists.

## Introduction

Evidence from biodiversity–ecosystem function (BEF hereafter) experiments during the past 30 years generally show positive relationships between biodiversity and productivity, soil carbon storage,

decomposition rates, and other ecosystem functions in experimental communities, revealing the importance of biodiversity in maintaining ecosystem functioning (*Tilman et al., 2012*; *van der Plas, 2019*). When research expands from experiments to natural systems, however, BEF relationships remain unclear in the natural assembled communities, with significant context dependency (*Hagan et al., 2021*; *van der Plas, 2019*; but see *Duffy et al., 2017*). One of the main reasons for these differences is because the landscape context surrounding natural communities regulates BEF (*Gonzalez et al., 2020*; *Liu et al., 2018*). Consideration of the impacts of landscape context on surrounding communities is key to understand complex BEF relationships in natural systems.

Human activities have modified natural ecosystems globally, with fragmented landscape contexts becoming increasingly widespread around the world (*Chase et al., 2020*; *Maxwell et al., 2016*). For example, at least 1.5 billion hectares of natural habitats on Earth have been converted to human-modified land since 2014, breaking apart continuous habitat into smaller and isolated fragments (*IPBES, 2019*). Fragmented landscape context typically involves different processes, such as habitat loss, that is, reducing habitat amount in the landscape, and fragmentation per se, that is, breaking apart of habitat for a given habitat amount in the landscape, including decreased mean size of habitat patches, increased number of habitat patches, increased isolation among habitat patches, etc. (*Fahrig, 2003*; *Ibáñez et al., 2014*; *Wang et al., 2014*). Habitat loss is often considered the major near-term threat to the biodiversity of terrestrial ecosystems (*Chase et al., 2020*; *Haddad et al., 2015*), while the impact of fragmentation per se remains debated (*Fletcher et al., 2023*; *Miller-Rushing et al., 2019*). Thus, habitat loss and fragmentation per se may have inconsistent ecological consequences and should be considered simultaneously to establish effective conservation strategies in fragmented landscapes (*Fahrig et al., 2019*; *Fletcher et al., 2018*; *Miller-Rushing et al., 2019*).

Fragmented landscape context can also affect BEF relationships. Previous studies have found that the magnitude and direction of BEF relationships vary across fragmented landscapes, including positive, negative, and non-significant relationships (*Godbold et al., 2011*; *Hagan et al., 2021*; *Rolo et al., 2018*; *Zirbel et al., 2019*). However, few studies have been conducted on how habitat loss and fragmentation per se regulate BEF relationships (*Liu et al., 2018*; *Wilson et al., 2016*). This lack of studies hampers our understanding of complex BEF relationships in fragmented natural ecosystems. In theory, habitat loss and fragmentation per se can regulate ecosystem function and the BEF relationship by altering species composition, interactions, and spatial asynchrony regardless of changes in species richness (*Liu et al., 2018*; *Thompson and Gonzalez, 2016*; *Tscharntke et al., 2012*). This is because species in communities are not ecologically equivalent and may respond differently to habitat loss and fragmentation per se, and contribute unequally to ecosystem function (*Devictor et al., 2008*; *Wardle and Zackrisson, 2005*). Therefore, considering changes in species composition can help to understand how habitat loss and fragmentation per se regulate BEF relationships.

The degree of habitat specialisation is a key ecological characteristic determining responses of species to habitat loss and fragmentation per se and the slope of BEF relationships (*Clavel et al., 2011*; *Gravel et al., 2011*). In fragmented landscapes, species with low levels of habitat specialisation (generalists) can use resources from different land covers, including focal habitat and non-habitat matrix, and thus are not sensitive to habitat loss and fragmentation per se (*Matthews et al., 2014*). Conversely, species with high levels of habitat specialisation (specialists) depend highly on resources in specific habitats, and thus are vulnerable to adverse effects from habitat loss and fragmentation per se (*Matthews et al., 2014*). In communities, specialists with specialised niches in resource use may contribute complementary roles to ecosystem functioning, whereas generalists with unspecialised niches in resource use may contribute redundant roles to ecosystem functioning due to overlapping niches (*Dehling et al., 2021*; *Denelle et al., 2020*; *Gravel et al., 2011*; *Wilsey et al., 2024*). Therefore, communities composed of specialists should have a higher niche complementarity effect in maintaining ecosystem functions and a more significant BEF relationship than communities composed of generalists. Habitat loss and fragmentation per se are often predicted to decrease the degree of habitat specialisation (the replacement of specialists in communities by generalists), possibly resulting in functional homogenisation of communities and reduced BEF relationships (*Clavel et al., 2011*; *Matthews, 2021*). However, few studies have evaluated this process in fragmented landscapes.

Currently, research on habitat loss and fragmentation per se focusses primarily on forest ecosystems (*Fardila et al., 2017*; *Haddad et al., 2015*; *Ma et al., 2023*). Grasslands have received considerably less attention, despite being one of the largest terrestrial ecosystems, and suffering severe

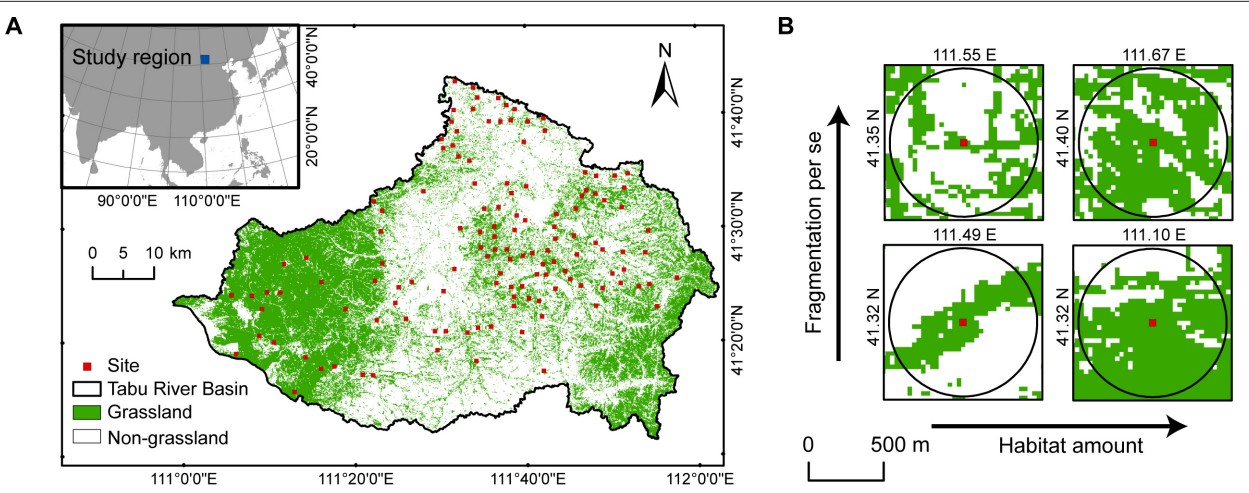

**Figure 1.** Map of study area in the Tabu River Basin, Inner Mongolia Autonomous Region, northern China. (**A**) Location of the 130 survey sites in the study area. (**B**) Examples of four survey sites with varying levels of habitat loss and fragmentation per se shown with a 500m radius buffer.

The online version of this article includes the following figure supplement(s) for figure 1:

**Figure supplement 1.** Biplot of the principal component analysis for calculating the single fragmentation index.

**Figure supplement 2.** Scatterplot of habitat amount and fragmentation for stratified sampling.

fragmentation due to human activities, such as agricultural reclamation and urbanisation (*Fardila et al., 2017*). The agro-pastoral ecotone of northern China is a typical anthropogenically fragmented grassland landscape caused by historical agricultural reclamation, especially in the late Qing Dynasty (about 1840–1912). Due to land policy reforms, the region has experienced a rapid expansion of farmland since the 1960s, converting continuous natural grasslands into smaller and isolated fragments, seriously threatening the conservation of BEFs (*Yan et al., 2022*; *Yan et al., 2023*; *Yang et al., 2020*). Based on 130 landscapes with different fragmentation levels in the agro-pastoral ecotone of northern China (*Figure 1*), we investigated how fragmented landscape context (habitat loss and fragmentation per se) impact the relationship between grassland plant diversity and above-ground productivity in the community. Specifically, we aimed to evaluate whether habitat loss and fragmentation per se would weaken the positive relationship between grassland plant diversity and above-ground productivity by reducing the habitat specialisation of the community.

## Results

### Relationship of habitat loss and fragmentation per se with grassland plant richness and above-ground biomass

A total of 130 vascular plant species were identified in our study sites, including 91 grassland specialists and 39 weeds (*Supplementary file 1*). Habitat loss was significantly negatively correlated with overall species richness ($R = -0.21$, $p < 0.05$, *Figure 2A*) and grassland specialist richness ($R = -0.41$, $p < 0.01$, *Figure 2A*), but positively correlated with weed richness ($R = 0.31$, $p < 0.01$, *Figure 2A*). Fragmentation per se was not significantly correlated with overall species richness and grassland specialist richness, but was significantly positively correlated with weed richness ($R = 0.26$, $p < 0.01$, *Figure 2B*). Habitat loss ($R = -0.39$, $p < 0.01$, *Figure 2C*) and fragmentation per se ($R = -0.26$, $p < 0.01$, *Figure 2D*) were both significantly negatively correlated with above-ground biomass.

### The relative effects of landscape context, plant diversity, and environmental factors on above-ground biomass

Results of the multi-model averaging for the four optimal models affecting above-ground biomass (*Supplementary file 2*) showed that plant diversity had the strongest relative effects on above-ground biomass than landscape context and environmental factors. Grassland specialist richness (estimate: 0.61, $p < 0.01$, *Figure 3*) and weed richness (estimate: 0.24, $p < 0.01$, *Figure 3*) had significant

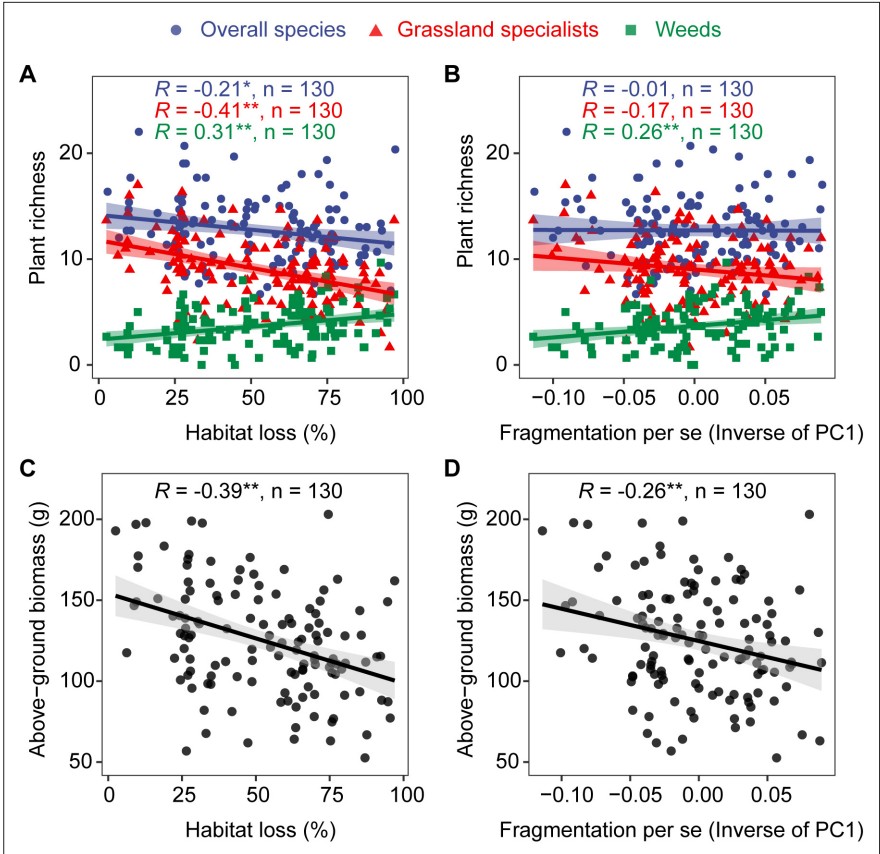

**Figure 2.** Correlation of habitat loss and fragmentation per se with grassland plant richness and above-ground biomass from 130 landscapes in the Tabu River Basin, a typical agro-pastoral ecotone of northern China. (**A**) Habitat loss and plant richness, (**B**) fragmentation per se and plant richness, (**C**) habitat loss and above-ground biomass, and (**D**) fragmentation per se and above-ground biomass. The R value in each panel is from the Pearson correlation coefficient analyses. The n in each panel is the number of surveying sites used in the Pearson correlation analyses. The trend lines in the figure are from linear regression models. The shaded area around the trend line represents the 95% confidence interval. * and ** represent significance at the 0.05 and 0.01 levels, respectively.

positive relative effects on above-ground biomass. Habitat loss (estimate: −0.16, p > 0.05, *Figure 3*) and fragmentation per se (estimate: −0.05, p > 0.05, *Figure 3*) had insignificant negative relative effects on above-ground biomass. Soil water content (SWC) had a significant positive relative effect on above-ground biomass (estimate: 0.19, p < 0.01, *Figure 3*), and land surface temperature (LST) had an insignificant negative relative effect on above-ground biomass (estimate: −0.01, p > 0.05, *Figure 3*).

## The impact of habitat loss and fragmentation per se on the relationship between grassland plant richness and above-ground biomass

The linear regression models showed that habitat loss had a significant negative modulating effect on the positive relationship between plant richness and above-ground biomass (estimate = −0.23, p < 0.05, *Supplementary file 3*), and fragmentation per se had no significant modulating effect (estimate = −0.10, p > 0.05, *Supplementary file 3*). The positive relationship between plant richness and above-ground biomass weakened with increasing levels of habitat loss, strengthened and then weakened with increasing levels of fragmentation per se (*Figure 4*).

The Fisher's C statistic indicated that the piecewise structural equation model fitted the data well (Fisher's C = 19.3, p-value >0.05, *Figure 5*). The piecewise structural equation model showed that the percentage of grassland specialists increased the positive effect of plant richness on above-ground biomass (path coefficient: 0.34, *Figure 5*). Habitat loss decreased the positive effect of plant richness on above-ground biomass by decreasing the percentage of grassland specialists (path coefficient:

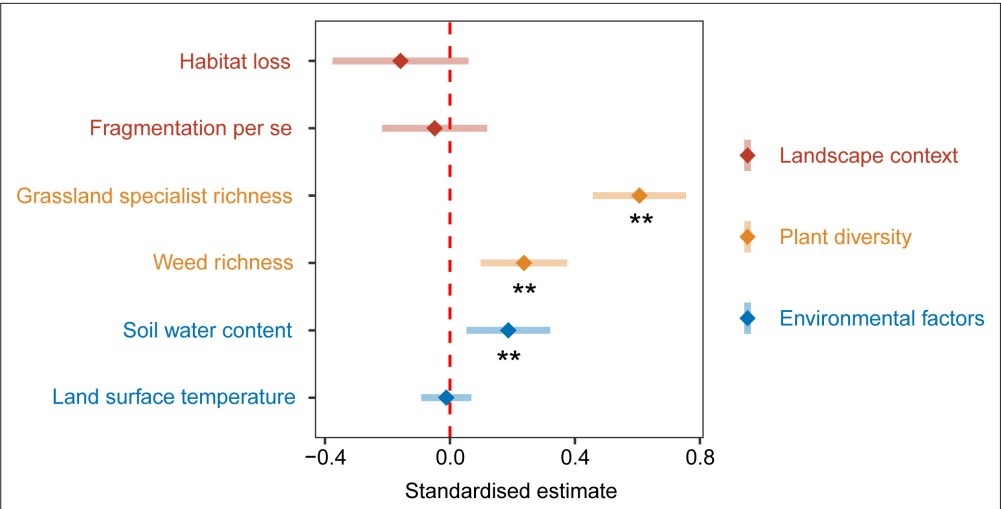

**Figure 3.** Standardised parameter estimates and 95% confidence intervals for landscape context, plant diversity, and environmental factors affecting above-ground biomass from 130 landscapes in the Tabu River Basin, a typical agro-pastoral ecotone of northern China. Standardised estimates and 95% confidence intervals are calculated by the multi-model-averaging method based on the four optimal models affecting above-ground biomass (*Supplementary file 2*). ** represent significance at the 0.01 level.

−0.46, *Figure 5*). Fragmentation per se had no significant effect on the percentage of grassland specialists.

Meanwhile, the piecewise structural equation model showed habitat loss to have an indirect negative effect on above-ground biomass through decreasing plant richness (path coefficient: −0.39, *Figure 5*) and through increasing fragmentation per se (path coefficient: 0.69, *Figure 5*) and in turn decreasing SWC (path coefficient: −0.35, *Figure 5*). Fragmentation per se had an indirect negative effect on above-ground biomass through decreasing SWC (path coefficient: −0.35, *Figure 5*) and an indirect positive effect on above-ground biomass through increasing plant richness (path coefficient: 0.26, *Figure 5*).

## Discussion
### Habitat loss and fragmentation per se had inconsistent effects on grassland plant diversity and ecosystem function

Although habitat loss and fragmentation per se are generally highly associated in natural landscapes, they are distinct ecological processes that determine decisions on effective conservation strategies (*Fahrig, 2017*; *Valente et al., 2023*). Our study evaluated the effects of habitat loss and fragmentation per se on grassland plant diversity and above-ground productivity in the context of fragmented landscapes in the agro-pastoral ecotone of northern China, with our results showing the effects of these two facets to not be consistent.

Consistent with previous studies, we found habitat loss significantly reduced grassland plant diversity, suggesting that habitat loss is a major threat to the biodiversity conservation of fragmented landscapes in this region (*Chase et al., 2020*; *Haddad et al., 2015*; *Ibáñez et al., 2014*). While for fragmentation per se, we found a positive effect on grassland plant diversity, in accordance with some recent evidence, the effects of fragmentation per se on biodiversity are more likely positive than negative for a given amount of habitat (*Gestich et al., 2022*; *Palmeirim et al., 2019*; *Riva and Fahrig, 2023*). Our study suggests that biodiversity conservation strategies for fragmented landscapes should consider optimising the habitat configuration in the landscape in addition to preventing habitat loss, such as increasing the number of habitat patches (*Arroyo-Rodríguez et al., 2020*; *Fahrig, 2017*). However, it is important to stress that the observed positive effect of fragmentation per se does not imply that increasing the isolation of grassland patches would promote biodiversity, as the metric of fragmentation per se used in our study was more related to patch density, edge density, and mean

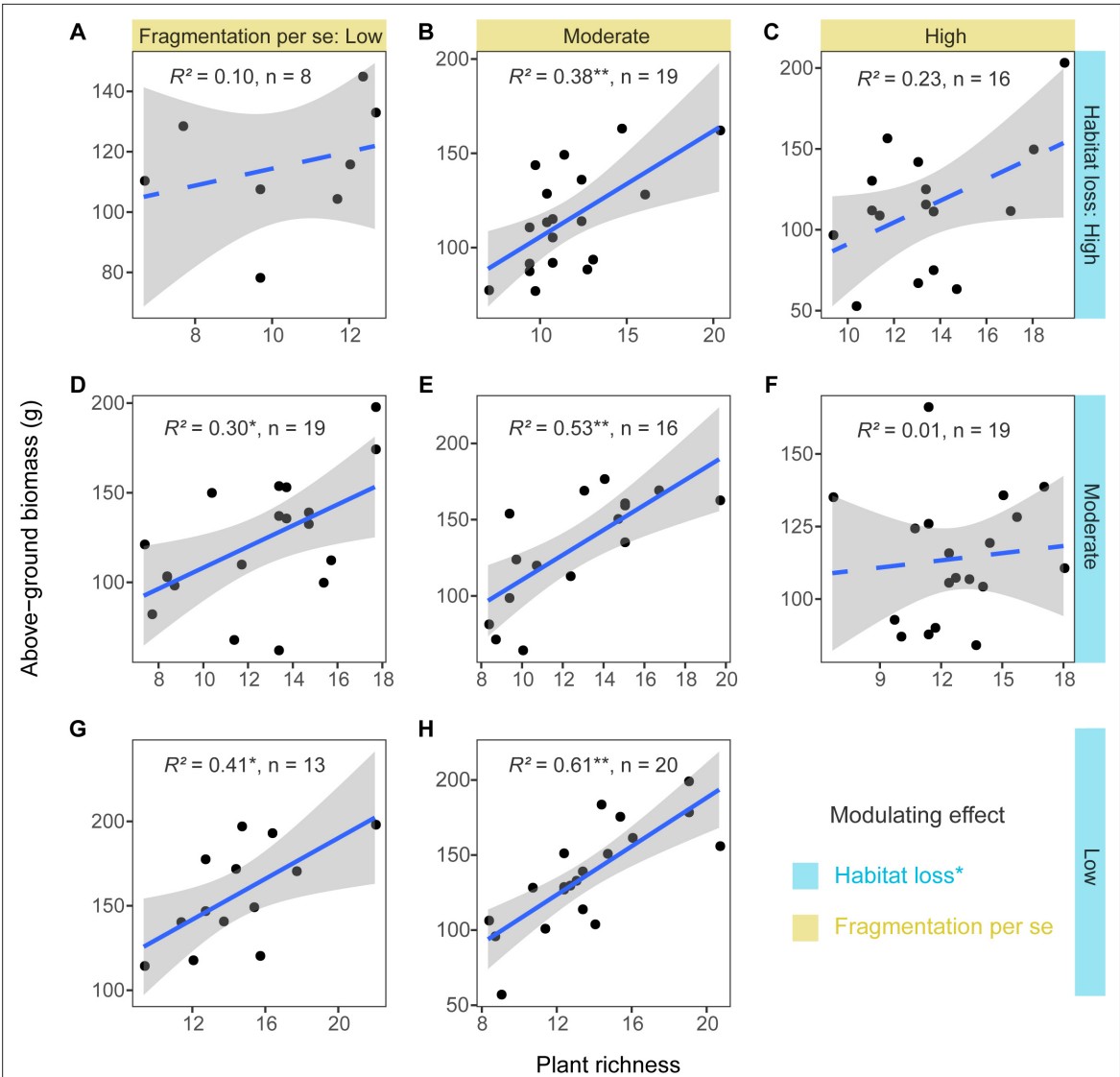

**Figure 4.** Relationships between grassland plant richness and above-ground biomass at different levels of habitat loss and fragmentation per se from 130 landscapes in the Tabu River Basin, a typical agro-pastoral ecotone of northern China. (**A**) High habitat loss and low fragmentation per se, (**B**) high habitat loss and moderate fragmentation per se, (**C**) high habitat loss and high fragmentation per se, (**D**) moderate habitat loss and low fragmentation per se, (**E**) moderate habitat loss and moderate fragmentation per se, (**F**) moderate habitat loss and high fragmentation per se, (**G**) low habitat loss and low fragmentation per se, and (**H**) low habitat loss and moderate fragmentation per se. The modulating effect in the figure represents the significance of interaction terms between habitat loss and fragmentation per se and plant richness for affecting above-ground biomass (*Supplementary file 3*). The $R^2$ values in each panel are from linear regression models. The *n* in each panel is the number of surveying sites used in the linear regression models. The blue solid and dashed trend lines represent the significant and not significant effects, respectively. The shaded area around the trend line represents the 95% confidence interval. * represent significance at the 0.05 level. ** represent significance at the 0.01 level.

patch area while relatively less related to patch isolation (*Supplementary file 4*). The potential threats from isolation still needs to be carefully considered in the conservation of biodiversity in fragmented landscapes (*Haddad et al., 2015*).

Our results showed for a metric of ecosystem function that landscape context affected above-ground productivity indirectly by influencing plant diversity and environment factors, which is consistent with the findings of previous studies in fragmented landscapes (*Allan et al., 2015*; *Rippel et al., 2020*). And we found fragmentation per se mediated the effect of habitat loss on environmental factors. A possible reason for this finding is fragmentation per se could directly alter environmental factors through enhanced edge effects (*Laurance et al., 2011*; *Smith et al., 2018*). In our study, fragmentation per se could lead to ecosystem drought and thus limit above-ground productivity in

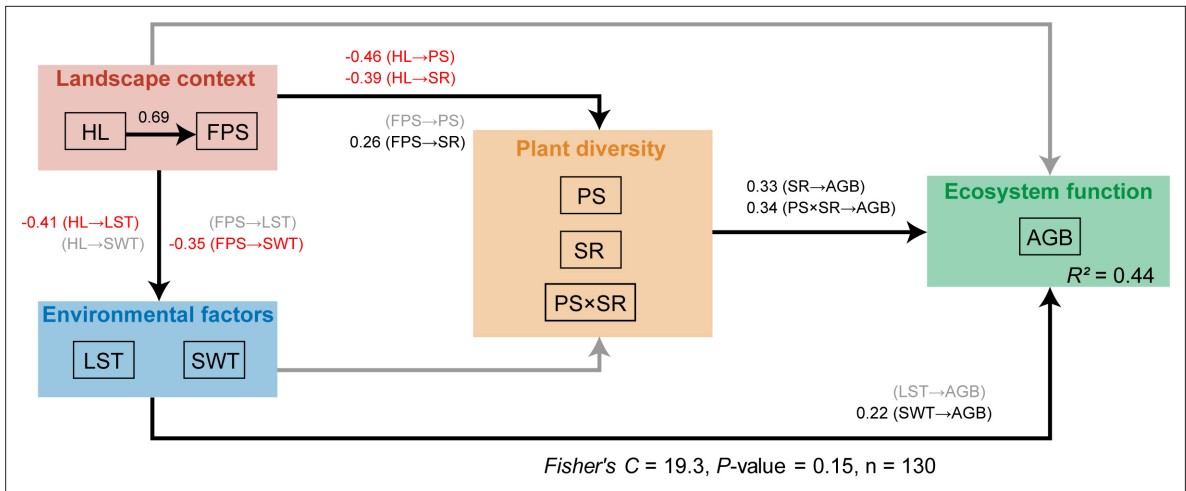

**Figure 5.** Results from the piecewise structural equation model with hypothesised paths showing how habitat loss and fragmentation per se alters the relationship between grassland plant richness and above-ground biomass from 130 landscapes in the Tabu River Basin, a typical agro-pastoral ecotone of northern China. HL: habitat loss; FPS: fragmentation per se; LST: land surface temperature; SWT: soil water content; PS: percentage of grassland specialists; SR: plant richness; PS × SR: interaction term between percentage of grassland specialists and plant richness; AGB: above-ground biomass. Black and grey solid arrows represent the significant and not significant effects at the 0.05 level, respectively. Black and red numbers on the solid arrows represent the significant positive and negative path coefficients, respectively. The Fisher's *C* and p-values are from the piecewise structural equation model.

The online version of this article includes the following figure supplement(s) for figure 5:

**Figure supplement 1.** Hypothetical model of habitat loss and fragmentation per se for the relationship between grassland plant richness and above-ground biomass affecting the percentage of grassland specialists in the community from 130 landscapes in the Tabu River Basin, a typical agro-pastoral ecotone of northern China.

grassland ecosystems. This is mainly because enhanced edge effects due to fragmentation per se could lead to greater desiccation and evapotranspiration rates in remaining habitats (*Smith et al., 2018*; *Tuff et al., 2016*). Therefore, our study suggests that habitat loss and fragmentation per se have inconsistent consequences on plant diversity and ecosystem function, which should be considered explicitly in developing conservation strategies in fragmented landscapes.

In our study, a possible mechanism for the positive impacts of fragmentation per se on plant diversity and above-ground productivity (indirect positive impact via plant diversity) is that fragmentation per se increases the habitat heterogeneity in the landscape, which can promote biodiversity through spatial asynchrony and spatial insurance effects (*Tscharntke et al., 2012*). Previous studies indicated that heterogeneity typically has nonlinear effects on BEF, as moderate heterogeneity can maximise spatial asynchrony (*Redon et al., 2014*; *Wilcox et al., 2017*). However, our study did not observe nonlinear patterns between fragmentation per se and plant diversity and above-ground productivity. This may be due to the low spatial heterogeneity of this area as a result of agricultural intensification (*Benton et al., 2003*; *Chen et al., 2019*). The gradient of fragmentation per se in our study may not cover the optimal heterogeneity levels for maximising plant diversity and above-ground productivity (*Thompson and Gonzalez, 2016*).

### Habitat loss rather than fragmentation per se weakened the magnitude of the positive relationship between plant diversity and ecosystem function

Understanding the direction and magnitude of BEF relationships in fragmented landscapes is essential to understanding the importance of biodiversity for ecosystem function in the changing world (*Gonzalez et al., 2020*; *van der Plas, 2019*). In naturally assembled communities, ecosystem functions may be dominated by complex environmental factors and landscape context, showing a weak or even negative correlation with biodiversity (*Grace et al., 2007*; *Hagan et al., 2021*; *Zirbel et al., 2019*). Our study found grassland plant diversity showed a stronger positive impact on above-ground productivity than landscape context and environmental factors. This result is consistent with findings

by *Duffy et al., 2017* in natural ecosystems, indicating grassland plant diversity has an important role in maintaining grassland ecosystem functions in the fragmented landscapes of the agro-pastoral ecotone of northern China.

Consistent with theoretical model predictions (*Liu et al., 2018*), our study found that fragmented landscape context significantly modulated the BEF relationship. We further found that habitat loss weakened the magnitude of the BEF relationship through decreasing the percentage of specialists in grassland communities, as specialists were more vulnerable to the negative effect of habitat loss and more associated with above-ground productivity than generalists. These findings indicate reducing the degree of habitat specialisation may be the mechanism of fragmented landscape context weakening the BEF relationship in this region (*Clavel et al., 2011*; *Gravel et al., 2011*). Meanwhile, our study demonstrates that habitat loss, rather than fragmentation per se, can decrease the degree of habitat specialisation by leading to the replacement of specialists by generalists in the community, thus weakening the BEF relationship. This is mainly because fragmentation per se did not decrease the grassland specialist richness in this region, whereas habitat loss decreased the grassland specialist richness and led to the invasion of more weeds from the surrounding farmland into the grassland community (*Yan et al., 2022*; *Yan et al., 2023*). Our findings suggest further expansion of farmland in the agro-pastoral ecotone of northern China would decrease the grassland plant diversity and its importance to above-ground productivity.

In addition, our study found that the BEF relationship showed a nonlinear pattern with increasing levels of fragmentation per se. For a given level of habitat loss, the positive BEF relationship was strongest at moderate fragmentation per se level and became neutral at high fragmentation per se level. This can be explained by the increased spatial asynchrony at moderate fragmentation per se level, which can promote niche complementary among species in the community and thus strengthen the BEF relationship (*Gonzalez et al., 2020*; *Thompson and Gonzalez, 2016*; *Tscharntke et al., 2012*). The neutral BEF relationship at high fragmentation per se level may be due to edge effects enhancing environmental filtering, thereby leading to functional redundancy among species and decoupling the BEF relationship (*Fetzer et al., 2015*; *Hu et al., 2016*; *Zambrano et al., 2019*). However, a recent study by *Hertzog et al., 2019* on temperate forest ecosystems found that the positive relationship between plant diversity and ecosystem multifunctionality strengthened with increasing forest fragmentation and peaked at the high level of forest fragmentation. This inconsistency can be explained by trade-offs between different ecosystem functions that may differ in their response to fragmentation per se (*Banks-Leite et al., 2020*). Therefore, future studies are needed to focus on multiple ecosystem functions, such as below-ground productivity, litter decomposition, and soil carbon stocks.

## Materials and methods
### Study area
Our study area is in the agro-pastoral ecotone of northern China, the Tabu River Basin in Siziwang Banner, Inner Mongolia Autonomous Region. The mean annual temperature ranges from 1.5 to 5.0°C, and the mean annual precipitation ranges from 225 to 322 mm. The type of soil is light chestnut soil. This area is a typical fragmented grassland landscape caused by agricultural intensification. Grassland is the dominant natural habitat type in this area, accounting for about 40.8% of the total area, with the dominant plant species being *Stipa krylovii* and S. *breviflora*. Farmland is the dominant matrix type in this area, accounting for about 30.6% of the total area, with the main crops grown being potatoes and maize. Further background information about the study area is described in our previous papers (*Yan et al., 2022*; *Yan et al., 2023*; *Yan et al., 2021*; *Zhang et al., 2021*).

### Sampling landscape selection
We quantified landscape-scale habitat loss and fragmentation per se in the study area to determine the spatial gradient of landscape context, then established the sampling landscapes. Grassland was defined as the focal habitat. Habitat amount was represented by the percentage of grassland cover in the landscape. Habitat loss was represented by the loss of grassland amount in the landscape. As the remaining grassland fragments in this region were mainly caused by grassland loss due to human activities such as cropland expansion (*Chen et al., 2019*; *Yang et al., 2020*), the percentage of non-grassland cover in the landscape was used in our study to represent habitat loss. Fragmentation per se

was estimated by calculating four landscape indices that reflect the different fragmentation processes for a given amount of habitat in the landscape (*Fahrig, 2003*; *Fahrig, 2017*): (1) patch density metric, representing an increase in the number of grassland patches in the landscape; (2) edge density metric, representing an increase in the grassland edges in the landscape; (3) mean patch area metric, representing a decrease in the mean size of grassland patches in the landscape; and (4) mean nearest-neighbour distance metric, representing an increase in the isolation among grassland patches in the landscape. The patch density metric reflects the breaking apart of habitat in the landscape, which is a direct reflection of the definition of fragmentation per se (*Fahrig et al., 2019*). The edge density metric reflects the magnitude of the edge effect caused by fragmentation (*Fahrig, 2017*). The mean patch area metric and the mean nearest-neighbour distance metric are associated with the area and distance effects of island biogeography, respectively, reflecting the processes of local extinction and dispersal of species in the landscape (*Fletcher et al., 2018*).

Given that habitat amount (or habitat loss) and fragmentation per se are typically highly correlated in natural landscapes, it is hard to disentangle their relative effects (*Fahrig, 2017*; *Smith et al., 2009*). We therefore used a quasi-experimental method to select sampling landscapes across the relative independent spatial gradient of habitat amount and fragmentation per se, which can reduce collinearity between them (*Butsic et al., 2017*; *Pasher et al., 2013*; *Reynolds et al., 2018*). To do so, we first used the moving window method (window size: 500 m radius buffer) to quantify grassland amount and the four landscape indices (patch density, edge density, mean patch area, and mean nearest-neighbour distance metric) surrounding all grassland cells. The 500 m radius buffer was used because our previous studies showed this buffer includes the optimal scale of spatial processes influencing grassland plant diversity in this region (*Yan et al., 2022*; *Yan et al., 2023*; *Zhang et al., 2021*). Second, to quantify the fragmentation per se level, we derived the first principal component (PC1) of the four landscape indices (*Hertzog et al., 2019*; *Rolo et al., 2018*). We took the inverse of the PC1 as a single fragmentation per se index (*Figure 1—figure supplement 1*), which was positively correlated with patch density, edge density, mean nearest-neighbour distance metric, and negatively with mean patch area (*Supplementary file 4*).

Based on the quartiles of grassland amount and single fragmentation per se index, we ranked these grassland cells into nine types: high–high, high–moderate, high–low, moderate–high, moderate–moderate, moderate–low, low–high, low–moderate, low–low grassland amount, and fragmentation per se (*Figure 1—figure supplement 2*). Given landscapes with a high grassland amount and high fragmentation are scarce in this region (*Figure 1—figure supplement 2*), we did not consider this type of landscape. Finally, we selected at least 20 grassland landscapes with a minimum distance condition using stratified sampling from each of the remaining eight grassland types as alternative sites for field surveys. The minimum distance between each landscape was at least 1000 m to prevent overlapping landscapes and potential spatial autocorrelation.

The land-cover data used to quantify grassland fragmentation were obtained via supervised classification on a cloud-free Landsat 8 TOA composite image (30 m resolution) from 2019 (*Yan et al., 2022*). We used the random forest classifier in the Google Earth Engine platform (*Gorelick et al., 2017*) for the supervised classification. The overall classification accuracy was 84.3 %, and the kappa coefficient was 0.81. The moving window analysis and all landscape metric calculations were performed in FRAG-STATS v4.2.1 based on the eight-cell neighbourhood rule (*McGarigal et al., 2012*). The principal component analysis was performed in the R programming language v. 4.0.3 (*R Development Core Team, 2020*), and stratified sampling was conducted in ArcGIS v10.3.

## BEF surveys

Based on the alternative sites selected above, we established 130 sites (30 m × 30 m) between late July and mid-August 2020 in the Tabu River Basin in Siziwang Banner, Inner Mongolia Autonomous Region (*Figure 1*). The types of the 130 sites were: 20 high–moderate, 13 high–low, 19 moderate–high, 16 moderate–moderate, 19 moderate–low, 16 low–high, 19 low–moderate, 8 low–low habitat amount, and fragmentation per se. In order to exclude the impact of historical agricultural activities, the habitat type of the established sites was natural grasslands with regional vegetation characteristics. Each site was not abandoned agricultural land, and there was no sign of agricultural reclamation.

At the 10 m × 10 m centre of each site, we randomly set up three 1 m × 1 m plots in a flat topographic area to investigate grassland vascular plant diversity and above-ground productivity. Plant

diversity was obtained by recording the number of vascular plant species in each plot. The above-ground productivity was obtained by harvesting the above-ground biomass of the plants in each plot and drying biomass at 65°C to a constant weight. We also investigated two environmental factors related to water and temperature (SWC and LST) in each site to consider their potential impact on plant diversity and ecosystem function. The SWC (%) was obtained by measuring the weight of wet soil samples before and after oven-drying at 105°C to a constant weight. The wet soil samples were collected from three 30 cm deep cores using a 5-cm diameter soil auger within three plots of each site. The LST (°C) was extracted from MODIS Land Surface Temperature/Emissivity daily product (MOD11A1) using the Google Earth Engine platform. For each site, the average daily LST of late July to mid-August 2020 was used in the following data analysis.

As grassland is the dominant habitat type in the fragmented landscape and farmland is the dominant matrix type, the specialists and generalists in this study were grouped as grassland specialists, that is, species that occur only in grassland, and weeds, that is, species that occur in both grassland and farmland. The classification of grassland specialists and weeds in this study was based on our experience with plant surveys in this region, the List of Main Crop Weeds in China, and available information in the Flora of China (http://www.iplant.cn/frps).

## Data analysis

For each site, we calculated the mean vascular plant richness of the three 1 m × 1 m plots, representing the vascular plant diversity, and the mean above-ground biomass of the three 1 m × 1 m plots, representing the above-ground productivity. The mean vascular plant richness and the mean above-ground biomass were assessed to be normally distributed by a Shapiro–Wilk normality test. The degree of habitat specialisation was represented by the percentage of grassland specialists in the community. Landscape context includes habitat loss and fragmentation per se. Habitat loss was represented by the percentage of non-grassland cover in the landscape and fragmentation per se was represented by the inverse of the PC1 of the four landscape indices (mean grassland patch area, mean nearest-neighbour distance among grassland patch, grassland patch density, and grassland patch edge). Environmental factors were LST and SWC.

First, to investigate the overall pattern between landscape context and grassland vascular plant diversity and above-ground productivity in this region, we used scatter plots and Pearson correlation analyses to present the pairwise relationship of habitat loss and fragmentation per se with plant richness (including grassland specialist richness and weed richness) and above-ground biomass.

Second, to investigate the relative importance of landscape context, grassland vascular plant diversity, and environmental factors on above-ground productivity, we used the multi-model-averaging method based on the Akaike information criterion corrected (AICc) for a small sample size (*Harrison et al., 2018*). We first constructed linear regression models, including the response variable (above-ground biomass) and all combinations of the predictor variables (habitat loss, fragmentation per se, grassland specialist richness, weed richness, LST, and SWC). The models with the lowest AICc value and a difference of less than two ΔAICc from the lowest AICc value were selected as optimal models. We then calculated the model-averaged standardised parameter estimate based on the optimal models as the relative effect of each influencing factor on above-ground biomass (*Harrison et al., 2018*). Before the analysis, we calculated the variance inflation factors (VIF) for each predictor variable to assess multicollinearity. The VIF of all explanatory variables was less than four (*Supplementary file 5*), suggesting no significant multicollinearity in the analysis (*Carrara et al., 2015*; *Dormann et al., 2013*).

Finally, to investigate how landscape context impacts the relationship between grassland plant diversity and above-ground productivity, we first used linear regression models to evaluate relationship between plant richness and above-ground biomass at low, moderate, and high levels of habitat loss and fragmentation per se, respectively. We then assessed the significance of interaction terms between habitat loss and fragmentation per se and plant richness in the linear regression models to evaluate whether they modulate the relationship between plant richness and above-ground biomass. Furthermore, we used a piecewise structural equation model to investigate the specific pathways in which habitat loss and fragmentation per se modulate the relationship between plant richness and above-ground biomass.

To do so, we first constructed a hypothetical conceptual model (*Figure 5—figure supplement 1*). Based on previous studies, we hypothesised that habitat loss and fragmentation per se could decrease above-ground biomass directly and indirectly by affecting plant richness and environmental factors (*Allan et al., 2015*; *Haddad et al., 2015*; *Rolo et al., 2018*). As habitat loss is often the main cause of fragmentation per se in natural landscapes, we hypothesised a causal relationship between habitat loss and fragmentation per se (*Didham et al., 2012*). We also hypothesised that habitat loss and fragmentation per se could both decrease the impact of plant richness on above-ground biomass by decreasing the percentage of grassland specialists in the community (*Clavel et al., 2011*; *Gravel et al., 2011*; *Liu et al., 2018*). We included the interaction between the percentage of grassland specialists and the plant richness as predictor variables to quantify the modulating effect of the percentage of grassland specialists for the impact of plant richness on above-ground biomass. We tested the hypothetical conceptual model using a piecewise structural equation model (*Lefcheck, 2016*). The global fit of the model was evaluated using the Fisher's $C$ statistic, and the hypothetical pathways were evaluated by standardised path coefficients.

All data analyses were performed in the R programming language v. 4.0.3 (*R Development Core Team, 2020*), with the following functions and packages. The Pearson correlation analysis, linear regression model, and the Shapiro–Wilk normality test were conducted with 'cor', 'lm' and 'shapiro. test' functions of the stats package (*R Development Core Team, 2020*). The multi-model averaging was conducted with the 'dredge' and 'model.avg' functions of the MuMIn package (*Bartoń, 2020*). The VIF was calculated with the 'vif' function of the car package (*Fox and Weisberg, 2019*). The piecewise structural equation model was conducted and tested with the 'psem' function of the piece-wiseSEM package (*Lefcheck, 2016*).

## Acknowledgements

We are very grateful to Peng Han, Shuangshuang Zhang, Xiaoqian Gong, Fengshi Li, Nier Su, Luyao Liu, Linjie Yao, and Guofeng Shi for assistance with field work. This study was supported by the National Natural Science Foundation of China (32071582), the Inner Mongolia Autonomous Region Science and Technology Plan Project (2021GG0392), and the Inner Mongolia Autonomous Region Natural Science Foundation (2023ZD24). No permits were required for fieldwork in this research.

## Additional information

### Funding

| Funder | Grant reference number | Author |
|---|---|---|
| National Natural Science Foundation of China | 32071582 | Qing Zhang |
| Inner Mongolia Autonomous Region Science and Technology Plan Project | 2021GG0392 | Qing Zhang |
| Inner Mongolia Autonomous Region Natural Science Foundation | 2023ZD24 | Qing Zhang |

The funders had no role in study design, data collection, and interpretation, or the decision to submit the work for publication.

### Author contributions

Yongzhi Yan, Formal analysis, Investigation, Visualization, Methodology, Writing - original draft, Writing - review and editing; Scott Jarvie, Writing - original draft, Writing - review and editing; Qing Zhang, Funding acquisition, Methodology, Writing - review and editing

### Author ORCIDs

Yongzhi Yan http://orcid.org/0000-0001-8493-6698

Scott Jarvie http://orcid.org/0000-0002-0086-2351

Qing Zhang http://orcid.org/0000-0002-3489-1417

Reviewer #1 (Public Review): https://doi.org/10.7554/eLife.91193.3.sa1

Reviewer #2 (Public Review): https://doi.org/10.7554/eLife.91193.3.sa2

Reviewer #3 (Public Review): https://doi.org/10.7554/eLife.91193.3.sa3

Author Response https://doi.org/10.7554/eLife.91193.3.sa4

## Additional files

### Supplementary files

• Supplementary file 1. List of 130 species of vascular plants recorded across 130 sites in this study.

• Supplementary file 2. Four optimal models of landscape context, environment factors, and plant diversity affecting above-ground biomass.

• Supplementary file 3. Effects of interaction terms between habitat loss and fragmentation per se and plant richness on above-ground biomass.

• Supplementary file 4. Summary of the principal component analysis for the four fragmentation indices.

• Supplementary file 5. Variance inflation factors of predictor variables for above-ground biomass.

• MDAR checklist

• Source data 1. The site location, landscape context, environmental factors, plant richness, and above-ground biomass.

### Data availability

All data analysed during this study are provided in the *Source data 1*.

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
