## [Editor Report · eLife assessment]

This **important** study advances our understanding of how landscape context affects the relationship between grassland plant diversity and biomass. This study used very well-designed approaches to analyze complex ecological relationships in real-world landscapes and thus provides **compelling** evidence to support its findings. The work will be of interest to landscape ecologists and community ecologists.

---

## [Referee Report · Reviewer #1 (Public Review)]

This is a well-designed study that explores the BEF relationships in fragmented landscapes. Although there are massive studies on BEF relationships, most of them were conducted at local scales, few considered the impacts of landscape variables. This study used a large dataset to specifically address this question and found that habitat loss weakened the BEF relationships. Overall, this manuscript is clearly written and has important implications for BEF studies as well as for ecosystem restoration.

I have read the response letter provided by the authors and think they have done a great job in addressing my concerns.

---

## [Referee Report · Reviewer #2 (Public Review)]

Summary:

In this manuscript, Yan et al. assess the effect of two facets of habitat fragmentation (i.e., habitat loss and habitat fragmentation per se) on biodiversity, ecosystem function, and the biodiversity-ecosystem function (BEF) relationship in grasslands of an agro-pastoral ecotone landscape in northern China. The authors use a stratified random sampling to select 130 study sites located within 500 m - radius landscapes varying along gradients of habitat loss and habitat fragmentation per se. In these study sites, the authors measure grassland specialist and generalist plant richness via field surveys, as well as above-ground biomass by harvesting and dry-weighting the grass communities in each 3 x 1m2 plots of the 130 study sites. The authors find that habitat loss and fragmentation per se have different effects on biodiversity, ecosystem function and the BEF relationship: whereas habitat loss was associated with a decrease in plant richness, fragmentation per se was not; and whereas fragmentation per se was associated with a decrease in above-ground biomass, habitat loss was not. Finally, habitat loss, but not fragmentation per se was linked to a decrease in the magnitude of the positive biodiversity-ecosystem functioning relationship, via reducing the percentage of grassland specialists in the community.

Strengths:

This study by Yan et al. is an exceptionally well-designed, well-written, clear and concise study shedding light on a longstanding, important question in landscape ecology and biodiversity-ecosystem functioning research. Via a stratified random sampling approach (cf. also "quasi-experimental design" Butsic et al. 2017), Yan et al. create an ideal set of study sites, where habitat loss and habitat fragmentation per se (usually highly correlated) are decorrelated and hence, separate effects of each of these facets on biodiversity and ecosystem function can be assessed statistically in "real-world" (and not experimental, cf. Duffy et al. 2017) communities. The authors use adequate and well-described methods to investigate their questions. The findings of this study add important empirical evidence from real-world grassland ecosystems that help to advance our theoretical understanding on landscape-moderation of biodiversity effects and provide important guidelines for conservation management likewise. Also, all figures are well-designed and clear.

Weaknesses:

I found only a few minor issues, mostly unclear descriptions that have now been revised for more clarity.

---

## [Referee Report · Reviewer #3 (Public Review)]

Summary: The authors aim to solve how landscape context impacts the community BEF relationship. They found habitat loss and fragmentation per se have inconsistent effects on biodiversity and ecosystem function. And habitat loss rather than fragmentation per se can weaken the positive BEF relationship by decreasing the degree of habitat specialization of the community.

Strengths: The authors provide a good background, and they have a good grasp of habitat fragmentation and BEF literature.

A major strength of this study is separating the impacts of habitat loss and fragmentation per se using the convincing design selection of landscapes with different combinations of habitat amount and fragmentation per se.

Another strength is considering the role of specialists and generalists in shaping the BEF relationship.

Weaknesses:

No

In the revised manuscript, the authors have provided more detailed information about the ecological significance of these fragmentation indices. and also integrated two environmental factors related to water and temperature (soil water content and land surface temperature) into the data analysis to control their potential impact.

---

## [Author Response]

The following is the authors’ response to the original reviews.

**eLife assessment**
This important study enhances our understanding of the effects of landscape context on grassland plant diversity and biomass. Notably, the authors use a well-designed field sampling method to separate the effects of habitat loss and fragmentation per se. Most of the data and analyses provide solid support for the findings that habitat loss weakens the positive relationship between grassland plant richness and biomass.

Response: Thanks very much for organizing the review of the manuscript. We are grateful to you for the recognition. We have carefully analyzed all comments of the editors and reviewers and revised our manuscript to address them. All comments and recommendations are helpfully and constructive for improving our manuscript. We have described in detail our response to each of comment below.

In addition to the reviewers' assessments, we have the following comments on your paper.(1) Some of the results are not consistent between figures. The relationships between overall species richness and fragmentation per se are not consistent between Figs. 3 and 5. The relationships between aboveground biomass and habitat loss are not consistent between Figs. 4 and 5. How shall we interpret these inconsistent results?

Response: Thanks for your insightful comments. The reason for these inconsistencies is that the linear regression model did not take into account the complex causal relationships (including direct and indirect effects) among the different influencing factors. The results in Figures 3 and 4 just represent the pairwise relationship pattern and relative importance, respectively. The causal effects of habitat loss and fragmentation per se on plant richness and above-ground biomass should be interpreted based on the structural equation model results (Figure 6). We have revised the data analysis to clear these inconsistent results. Line 225-228

In the revised manuscript, we have added the interpretation for these inconsistent results. The inconsistent effects between Figures 3 and 6 suggest that fragmentation per se actually had a positive effect on plant richness after accounting for the effects of habitat loss and environmental factors simultaneously.

The inconsistent effects between Figures 4 and 6 are because the effects of habitat loss and fragmentation per se on above-ground biomass were mainly mediated by plant richness and environmental factors, which had no significant direct effect (Figure 6). Thus, habitat loss and fragmentation per se showed no significant relative effects on above-ground biomass after controlling the effects of plant richness and environmental factors (Figure 4).

(2) One of the fragmentation indices, mean patch area metric, seems to be more appropriate as a measure of habitat loss, because it represents "a decrease in grassland patch area in the landscape".

Response: Thanks for your insightful comments. We apologize for causing this confusion. The mean patch area metric in our study represents the mean size of grassland patches in the landscape for a given grassland amount. Previous studies have often used the mean patch metric as a measure of fragmentation, which can reflect the processes of local extinction in the landscape (Fahrig, 2003; Fletcher et al., 2018). We have revised the definition of the mean patch area metric and added its ecological implication in the revised manuscript to clarify this confusion.

(3) It is important to show both the mean and 95% CI (or standard error) of the slope coefficients regarding to Figs. 3 and 6.

Response: Thanks for your suggestions. We have added the 95% confidence intervals to the Figure 3 and Figure 6 in the revised manuscript.

(4) It would be great to clarify what patch-level and landscape-level studies are in lines 302-306. Note that this study assesses the effects of landscape context on patch-level variables (i.e., plot-based plant richness and plot-based grassland biomass) rather than landscape-level variables (i.e., the average or total amount of biomass in a landscape).

Response: Thanks for your insightful comment. We agree with your point that our study investigated the effect of fragmented landscape context (habitat loss and fragmentation per se) on plot-based plant richness and plot-based above-ground biomass rather than landscape-level variables.

Therefore, we no longer discussed the differences between the patch-level and landscape-level studies here, instead focusing on the different ecological impacts of habitat loss and fragmentation per se in the revised manuscript.

Line 369-374:

“Although habitat loss and fragmentation per se are generally highly associated in natural landscapes, they are distinct ecological processes that determine decisions on effective conservation strategies (Fahrig, 2017; Valente et al., 2023). Our study evaluated the effects of habitat loss and fragmentation per se on grassland plant diversity and above-ground productivity in the context of fragmented landscapes in the agro-pastoral ecotone of northern China, with our results showing the effects of these two facets to not be consistent.”

(5) One possible way to avoid the confusion between "habitat fragmentation" and "fragmentation per se" could be to say "habitat loss and fragmentation per se" when you intend to express "habitat fragmentation".

Response: Thanks for your constructive suggestions. To avoid this confusion, we no longer mention habitat fragmentation in the revised manuscript but instead express it as habitat loss and fragmentation per se.

**Reviewer #1 (Public Review):**
This is a well-designed study that explores the BEF relationships in fragmented landscapes. Although there are massive studies on BEF relationships, most of them were conducted at local scales, few considered the impacts of landscape variables. This study used a large dataset to specifically address this question and found that habitat loss weakened the BEF relationships. Overall, this manuscript is clearly written and has important implications for BEF studies as well as for ecosystem restoration.

Response: We are grateful to you for the recognition and constructive comments. All the comments and suggestions are very constructive for improving this manuscript. We have carefully revised the manuscript following your suggestions. All changes are marked in red font in the revised manuscript.

My only concern is that the authors should clearly define habitat loss and fragmentation. Habitat loss and fragmentation are often associated, but they are different terms. The authors consider habitat loss a component of habitat fragmentation, which is not reasonable. Please see my specific comments below.

Response: We agree with your point. In the revised manuscript, we no longer consider habitat loss and fragmentation per se as two facets of habitat fragmentation. We have clearly defined habitat loss and fragmentation per se and explicitly evaluated their relative effects on plant richness, above-ground biomass, and the BEF relationship.

**Reviewer #1 (Recommendations For The Authors):**
Title: It is more proper to say habitat loss, rather than habitat fragmentation.

Response: Thanks for your suggestion. We have revised the title to “Habitat loss weakens the positive relationship between grassland plant richness and above-ground biomass”

Line 22, remove "Anthropogenic", this paper is not specifically discussing habitat fragmentation driven by humans.

Response: Thanks for your suggestion. We have removed the “Anthropogenic” from this sentence.

Line 26, revise to "we investigated the effects of habitat loss and fragmentation per se on plant richness... in grassland communities by using a structural equation model".

Response: Thanks for your suggestion. We have revised this sentence.

Line 25-28:

“Based on 130 landscapes identified by a stratified random sampling in the agro-pastoral ecotone of northern China, we investigated the effects of landscape context (habitat loss and fragmentation per se) on plant richness, above-ground biomass, and the relationship between them in grassland communities using a structural equation model.”

Line 58-60, habitat fragmentation generally involves habitat loss, but habitat loss is independent of habitat fragmentation, it is not a facet of habitat fragmentation.

Response: Thanks for your insightful comment. We have no longer considered habitat loss and fragmentation per se as two facets of habitat fragmentation. In the revised manuscript, we consider habitat loss and fragmentation as two different processes in fragmented landscapes.

Line 65-67, this sentence is not very relevant to this paragraph and can be deleted.

Response: Thanks for your suggestion. We have deleted this sentence from the paragraph.

Line 87-90, these references are mainly based on microorganisms, are there any references based on plants? These references are more relevant to this study. In addition, this is a key mechanism mentioned in this study, this section needs to be strengthened with more evidence and further exploration.

Response: Thanks for your comment and suggestion. Thanks for your comment and suggestion. We have added some references based on plants here to strengthen the evidence and mechanism of habitat specialisation determines the BEF relationship.

Line 89-95:

“In communities, specialists with specialised niches in resource use may contribute complementary roles to ecosystem functioning, whereas generalists with unspecialised in resource use may contribute redundant roles to ecosystem functioning due to overlapping niches (Dehling et al., 2021; Denelle et al., 2020; Gravel et al., 2011; Wilsey et al., 2023). Therefore, communities composed of specialists should have a higher niche complementarity effect in maintaining ecosystem functions and a more significant BEF relationship than communities composed of generalists.”

Denelle, P., Violle, C., DivGrass, C., Munoz, F. 2020. Generalist plants are more competitive and more functionally similar to each other than specialist plants: insights from network analyses. Journal of Biogeography 47: 1922-1933.

Dehling, D.M., Bender, I.M.A., Blendinger, P.G., Böhning-Gaese, K., Muñoz, M.C., Neuschulz, E.L., Quitián, M., Saavedra, F., Santillán, V., Schleuning, M., Stouffer, D.B. 2021. Specialists and generalists fulfil important and complementary functional roles in ecological processes. Functional Ecology 35: 1810-1821.

Wilsey, B., Martin, L., Xu, X., Isbell, F., Polley, H.W. 2023. Biodiversity: Net primary productivity relationships are eliminated by invasive species dominance. Ecology Letters.

Line 129-130, Although you can use habitat loss in the discussion or the introduction, here preferably use habitat amount or habitat area, rather than habitat loss in this case. Habitat loss represents changes in habitat area, but the remaining grasslands could be the case of natural succession or other processes, rather than loss of natural habitat.

Response: Thanks for your insightful comment. We agree with your point. In the revised manuscript, we have explicitly stated that habitat loss was represented by the loss of grassland amount in the landscape.

Since the remaining grassland fragments in this region were mainly caused by grassland loss due to human activities such as cropland expansion (Chen et al., 2019; Yang et al., 2020), we used the percentage of non-grassland cover in the landscape to represent habitat loss in our study.

Line 132-135:

“Habitat loss was represented by the loss of grassland amount in the landscape. As the remaining grassland fragments in this region were mainly caused by grassland loss due to human activities such as cropland expansion (Chen et al., 2019; Yang et al., 2020), the percentage of non-grassland cover in the landscape was used in our study to represent habitat loss.”

Lines 245-246, please also give more details of the statistical results, such as n, r value et al in the text.

Response: Thanks for your suggestion. We have added the details of the statistical results in the revised manuscript.

Line 283-290:

“Habitat loss was significantly negatively correlated with overall species richness (R = -0.21, p < 0.05, Figure 3a) and grassland specialist richness (R = -0.41, p < 0.01, Figure 3a), but positively correlated with weed richness (R = 0.31, p < 0.01, Figure 3a). Fragmentation per se was not significantly correlated with overall species richness and grassland specialist richness, but was significantly positively correlated with weed richness (R = 0.26, p < 0.01, Figure 3b). Habitat loss (R = -0.39, p < 0.01, Figure 3c) and fragmentation per se (R = -0.26, p < 0.01, Figure 3d) were both significantly negatively correlated with above-ground biomass.”

Fig. 5, is there any relationship between habitat amount and fragmentation per se in this study?

Response: Thanks for your insightful comment. We have considered a causal relationship between habitat loss and fragmentation per se in the structural equation model. We have discussed this relationship in the revised manuscript.

Line 290-293, how about the BEF relationships with different fragmentation levels? I may have missed something somewhere, but it was not shown here.

Response: Thanks for your insightful comment. We have added the BEF relationships with different fragmentation per se levels here.

Line 323-340:

“The linear regression models showed that habitat loss had a significant positive modulating effect on the positive relationship between plant richness and above-ground biomass, and fragmentation per se had no significant modulating effect (Figure 5). The positive relationship between plant richness and above-ground biomass weakened with increasing levels of habitat loss, strengthened and then weakened with increasing levels of fragmentation per se.

**Author response image 1. sa4fig1:** Relationships between grassland plant richness and above-ground biomass at different levels of habitat loss and fragmentation per se from 130 landscapes in the Tabu River Basin, a typical agro-pastoral ecotone of northern China. (a) High habitat loss and low fragmentation per se, (b) high habitat loss and moderate fragmentation per se, (c) high habitat loss and high fragmentation per se, (d) moderate habitat loss and low fragmentation per se, (e) moderate habitat loss and moderate fragmentation per se, (f) moderate habitat loss and high fragmentation per se, (g) low habitat loss and low fragmentation per se, (h) low habitat loss and moderate fragmentation per se. The R2 values in each panel are from linear regression models. The n in each panel is the number of surveying sites used in the linear regression models. The blue solid and dashed trend lines represent the significant and not significant effects, respectively. The shaded area around the trend line represents the 95% confidence interval. * represent significance at the 0.05 level. ** represent significance at the 0.01 level.

DiscussionThe Discussion (Section 4.2) needs to be revised and focused on your key findings, it is habitat loss, not fragmentation per se, that weakens the BEF relationships.

Response: Thanks for your insightful comment and suggestion. In the revised manuscript, we have rephrased the Discussion (Section 4.2) to mainly discuss the inconsistent effects of habitat loss and fragmentation per se on the BEF relationship.

Line 414-416:

“4.2 Habitat loss rather than fragmentation per se weakened the magnitude of the positive relationship between plant diversity and ecosystem function”

The R2 in the results are low (e.g., Fig. 3), please also mention other variables that might influence the observed pattern in the Discussion, such as soil and topography, though I understand it is difficult to collect such data in this study.

Response: Thanks for your insightful comment and suggestion. We agree with you and reviewer 3 that the impact of environmental factors should also be considered.

Therefore, we have considered two environmental factors related to water and temperature (soil water content and land surface temperature) in the analysis and discussed their impacts on plant diversity and above-ground biomass in the revised manuscript.

Lines 344-345, its relative importance was stronger in the intact landscape than that of the fragmented landscape?

Response: We apologize for making this confusion. We have rephrased this sentence.

Line 422-426:

“Our study found grassland plant diversity showed a stronger positive impact on above-ground productivity than landscape context and environmental factors. This result is consistent with findings by Duffy et al. (2017) in natural ecosystems, indicating grassland plant diversity has an important role in maintaining grassland ecosystem functions in the fragmented landscapes of the agro-pastoral ecotone of northern China.”

**Reviewer #2 (Public Review):**
Summary:In this manuscript, Yan et al. assess the effect of two facets of habitat fragmentation (i.e., habitat loss and habitat fragmentation per se) on biodiversity, ecosystem function, and the biodiversity-ecosystem function (BEF) relationship in grasslands of an agro-pastoral ecotone landscape in northern China. The authors use stratified random sampling to select 130 study sites located within 500m-radius landscapes varying along gradients of habitat loss and habitat fragmentation per se. In these study sites, the authors measure grassland specialist and generalist plant richness via field surveys, as well as above-ground biomass by harvesting and dry-weighting the grass communities in each 3 x 1m2 plots of the 130 study sites. The authors find that habitat loss and fragmentation per se have different effects on biodiversity, ecosystem function and the BEF relationship: whereas habitat loss was associated with a decrease in plant richness, fragmentation per se was not; and whereas fragmentation per se was associated with a decrease in above-ground biomass, habitat loss was not. Finally, habitat loss, but not fragmentation per se was linked to a decrease in the magnitude of the positive biodiversity-ecosystem functioning relationship, by reducing the percentage of grassland specialists in the community.Strengths:This study by Yan et al. is an exceptionally well-designed, well-written, clear and concise study shedding light on a longstanding, important question in landscape ecology and biodiversity-ecosystem functioning research. Via a stratified random sampling approach (cf. also "quasi-experimental design" Butsic et al. 2017), Yan et al. create an ideal set of study sites, where habitat loss and habitat fragmentation per se (usually highly correlated) are decorrelated and hence, separate effects of each of these facets on biodiversity and ecosystem function can be assessed statistically in "real-world" (and not experimental, cf. Duffy et al. 2017) communities. The authors use adequate and well-described methods to investigate their questions. The findings of this study add important empirical evidence from real-world grassland ecosystems that help to advance our theoretical understanding of landscape-moderation of biodiversity effects and provide important guidelines for conservation management.Weaknesses:I found only a few minor issues, mostly unclear descriptions in the study that could be revised for more clarity.

Response: Thanks very much for your review of the manuscript. We are grateful to you for the recognition. All the comments and suggestions are very insightful and constructive for improving this manuscript. We have carefully studied the literature you recommend and revised the manuscript carefully following your suggestions. All changes are marked in red font in the revised manuscript.

**Reviewer #2 (Recommendations For The Authors):**
Specific comments(1) Some aspects of the Methods section were not entirely clear to me, could you revise them for more clarity?(a) Whereas you describe 4 main facets of fragmentation per se that are used to create the PC1 as a measure of overall fragmentation per se, it looks as if this PC1 is mainly driven by 3 facets only (ED, PD and AREA_MN), and patch isolation (nearest neighbour distance, ENN) having a relatively low loading on PC1 (Figure A1). I think it would be good to discuss this fact and the consequences of it, that your definition of fragmentation is focused more on edge density, patch density and mean patch area, and less on patch isolation in your Discussion section?

Response: Thanks for your insightful comment and suggestion. We agree with your point. We have discussed this fact and its implications for understanding the effects of fragmentation per se in our study.

Line 384-389:

“However, it is important to stress that the observed positive effect of fragmentation per se does not imply that increasing the isolation of grassland patches would promote biodiversity, as the metric of fragmentation per se used in our study was more related to patch density, edge density and mean patch area while relatively less related to patch isolation (Appendix Table A1). The potential threats from isolation still need to be carefully considered in the conservation of biodiversity in fragmented landscapes (Haddad et al., 2015).”

(b) Also, from your PCA in Figure A1, it seems that positive values of PC1 mean "low fragmentation", whereas high values of PC1 mean "high fragmentation", however, in Figure A2, the inverse is shown (low values of PC1 = low fragmentation, high values of PC1 = high fragmentation). Could you clarify in the Methods section, if you scaled or normalized the PC1 to match this directionality?

Response: We apologize for making this confusion. In order to be consistent with the direction of change in fragmentation per se, we took the inverse of the PC1 as a single fragmentation per se index, which was positively correlated with patch density, edge density, mean nearest-neighbor distance metric, and negatively with mean patch area (Appendix Figure A1 and Table A1). We have clarified this point in the Method section.

Line 160-163:

“We took the inverse of the PC1 as a single fragmentation per se index, which was positively correlated with patch density, edge density, mean nearest-neighbor distance metric, and negatively with mean patch area (Appendix Figure A1 and Table A1).”

(c) On line 155 you describe that you selected at least 20 landscapes using stratified sampling from each of the eight groups of habitat amount and fragmentation combination. Could you clarify: (1) did you randomly sample within these groups with a minimum distance condition* or was it a non-random selection according to other criteria? *(I think you could move the "To prevent overlapping landscapes..." sentence up here to the description of the landscape selection process) (2) Why did you write "at least 20 landscapes" - were there in some cases more or less landscapes selected? 130 study landscapes divided by 8 groups only gives you 16.25, hence, at least for some groups there were less than 20 landscapes? Could you describe your final dataset in more detail, i.e. the number of landscapes per group and potential repercussions for your analysis?

Response: Thanks for your insightful comments. In the revised manuscript, we have rephrased the method to provide more detail for the sampling landscape selection.

(1) Line 169-172

We randomly selected at least 20 grassland landscapes with a minimum distance condition using stratified sampling from each of the remaining eight grassland types as alternative sites for field surveys. The minimum distance between each landscape was at least 1000 m to prevent overlapping landscapes and potential spatial autocorrelation.

(2) Line 184-191

The reason for selecting at least 20 grassland landscapes of each type in this study was to ensure enough alternative sites for the field survey. This is because the habitat type of some selected sites was not the natural grasslands, such as abandoned agricultural land. Some of the selected sites may not be permitted for field surveys.

Thus, we finally established 130 sites in the field survey. The types of the 130 sites were: 19 high-moderate, 14 high-low, 19 moderate-high, 16 moderate-moderate, 18 moderate-low, 16 low-high, 17 low-moderate, 11 low-low habitat amount and fragmentation per se.

(d) On line 166, you describe that you established 130 sites of 30 m by 30 m - I assume they were located (more or less) exactly in the centre of the selected 500 m - radius landscapes? Were they established so that they were fully covered with grassland? And more importantly, how did you establish the 10 m by 10 m areas and the 1 m2 plots within the 30 m by 30 m sites? Did you divide the 30 m by 30 m areas into three rectangles of 10 m by 10 m and then randomly established 1 m2 plots? Were the 1 m2 plots always fully covered with grassland/was there a minimum distance to edge criterion? Please describe with more detail how you established the 1 m2 study sites, and how many there were per landscape.

Response: Thanks for your insightful comments. In the revised manuscript, we have provided more detailed information on how to set up 130 sites of 30 m by 30 m and three plots of 1 m by 1 m.

(1) As these 130 sites were selected based on the calculation of the moving window, they were located (more or less) exactly in the centre of the 500-m radius buffer.

(2) These sites were fully covered with grassland because their size (30 m by 30 m) was the same as the size of the grassland cell (30 m by 30 m) used in the calculation of the moving window.

(3) We randomly set up three 1 m * 1 m plots in a flat topographic area at the 10 m * 10 m centre of each site. Thus, there was a minimum distance of 10 m to the edge for each 1 m * 1 m plot.

(4) There are three 1 m * 1 m plots per landscape.

Line 182-191:

“Based on the alternative sites selected above, we established 130 sites (30 m * 30 m) between late July to mid-August 2020 in the Tabu River Basin in Siziwang Banner, Inner Mongolia Autonomous Region (Figure 1). The types of the 130 sites were: 19 high-moderate, 14 high-low, 19 moderate-high, 16 moderate-moderate, 18 moderate-low, 16 low-high, 17 low-moderate, 11 low-low habitat amount and fragmentation per se. In order to exclude the impact of historical agricultural activities, the habitat type of the established sites was natural grasslands with regional vegetation characteristics. Each site was not abandoned agricultural land, and there was no sign of agricultural reclamation.

At the 10 m * 10 m center of each site, we randomly set up three 1 m * 1 m plots in a flat topographic area to investigate grassland vascular plant diversity and above-ground productivity.”

(e) Line 171: could you explain what you mean by reclaimed?

Response: Thanks for your comment. The “reclaimed” means that historical agricultural activities. We have rephrased this sentence to make it more explicit.

Line 186-189:

“In order to exclude the impact of historical agricultural activities, the habitat type of the established sites was natural grasslands with regional vegetation characteristics. Each site was not abandoned agricultural land, and there was no sign of agricultural reclamation.”

(f) Line 188 ff.: Hence your measure of productivity is average-above ground biomass per 1 m2. I think it would add clarity if you highlighted this more explicitly.

Response: Thanks for your suggestion. We have highlighted that the productivity in our study was the average above-ground biomass per 1 m * 1 m plots in each site.

Line 215-217:

“For each site, we calculated the mean vascular plant richness of the three 1 m * 1 m plots, representing the vascular plant diversity, and mean above-ground biomass of the three 1 m * 1 m plots, representing the above-ground productivity.”

(2) All figures are clear and well-designed!(a) Just as a suggestion: in Figures 3 and 6, you could maybe add the standard errors of the mean as well?

Response: Thanks for your suggestion. In the revised manuscript, we have added the standard errors of the mean in Figures 3 and 6.

(b) Figure 4: Could you please clarify: Which models were the optimal models on which these model-averaged standardized parameter estimates were based on? And hence, the optimal models contained all 4 predictors (otherwise, no standardized parameter estimate could be calculated)? Or do these model-averaged parameters take into account all possible models (and not only the optimal ones)?

Response: Thanks for your suggestion. We selected the four optimal models based on the AICc value to calculate the model-averaged standardized parameter estimates. The four optimal models contained all predictors in Figure 4. We have added the four optimal models in Appendix Table A3.

Appendix:

**Author response table 1. sa4table1:** Four optimal models of landscape context, environment factors, and plant diversity affecting above-ground biomass.

Ranking	Model	R^2^	AICc
1	AGB~HL+SWT+GSR+WR	0.47**	1213.7
2	AGB~HL+FPS+SWT+GSR+WR	0.47**	1215.2
3	AGB~HL+LST+SWT+GSR+WR	0.47**	1215.3
4	AGB~FPS+SWT+GSR+WR	0.46**	1215.3

Note: AGB: above-ground biomass; HL: habitat loss; FPS: fragmentation per se; SWT: soil water content; LST: land surface temperature; GSR: grassland specialist richness; WR: weed richness; **: significance at the 0.01 level.”.

(c) Please add in all Figures (i.e., Figures 4, 5 and 6, Figure 6 per "high, moderate and low-class") the number of study units the analyses were based on.

Response: Thanks for your suggestion. In the revised manuscript, we have added the number of study units the analyses were based on in all Figures.

(d) Figure 6: I think it would be more consistent to add a second plot where the BEF-relationship is shown for low, moderate and high levels of habitat fragmentation per se. Could you also add a clearer description in the Methods and/or Results section of how you assessed if habitat amount or fragmentation per se affected the BEF-relationship? I.e. based on the significance of the interaction term (habitat amount x species richness) in a linear model?

Response: Thanks for your insightful comment and suggestion. We have added a second plot in Figure 5 to show the BEF relationship at low, moderate and high levels of fragmentation per se.

Line 328-340:

**Author response image 2. sa4fig2:** Relationships between grassland plant richness and above-ground biomass at different levels of habitat loss and fragmentation per se from 130 landscapes in the Tabu River Basin, a typical agro-pastoral ecotone of northern China. (a) High habitat loss and low fragmentation per se, (b) high habitat loss and moderate fragmentation per se, (c) high habitat loss and high fragmentation per se, (d) moderate habitat loss and low fragmentation per se, (e) moderate habitat loss and moderate fragmentation per se, (f) moderate habitat loss and high fragmentation per se, (g) low habitat loss and low fragmentation per se, (h) low habitat loss and moderate fragmentation per se. The R2 values in each panel are from linear regression models. The n in each panel is the number of surveying sites used in the linear regression models. The blue solid and dashed trend lines represent the significant and not significant effects, respectively. The shaded area around the trend line represents the 95% confidence interval. * represent significance at the 0.05 level. ** represent significance at the 0.01 level.

We determined whether habitat loss and fragmentation per se moderated the BEF relationship by testing the significance of their interaction term with plant richness. We have added a clearer description in the Methods section of the revised manuscript.

Line 245-250:

“We then assessed the significance of interaction terms between habitat loss and fragmentation per se and plant richness in the linear regression models to evaluate whether they modulate the relationship between plant richness and above-ground biomass. Further, we used a piecewise structural equation model to investigate the specific pathways in which habitat loss and fragmentation per se modulate the relationship between plant richness and above-ground biomass.”

(3) While reading your manuscript, I missed a discussion on the potential non-linear effects of habitat amount and fragmentation per se. In your study, it seems that the effects of habitat amount and fragmentation per se on biodiversity and ecosystem function are quite linear, which contrasts previous research highlighting that intermediate levels of fragmentation/heterogeneity could maximise spatial asynchrony, biodiversity and ecosystem function (e.g. Redon et al. 2014, Thompson & Gonzalez 2016, Tscharntke et al. 2012, Wilcox et al. 2017). I think it would add depth to your study if you discussed your finding of linear effects of habitat amount and fragmentation on biodiversity, ecosystem functioning and BEF. For example:

Response: Thanks for your constructive suggestions. We have carefully studied the literature (e.g. Redon et al. 2014, Thompson & Gonzalez 2016, Tscharntke et al. 2012, Wilcox et al. 2017), which highlights that intermediate levels of fragmentation/heterogeneity could maximise spatial asynchrony, biodiversity and ecosystem function.

In the revised manuscript, we have added the discussion about the linear positive effects of fragmentation on plant diversity and above-ground productivity and discussed possible reasons for this linear effect.

Line 402-413:

“In our study, a possible mechanism for the positive impacts of fragmentation per se on plant diversity and above-ground productivity (indirect positive impact via plant diversity) is that fragmentation per se increases the habitat heterogeneity in the landscape, which can promote biodiversity through spatial asynchrony and spatial insurance effects (Tscharntke et al., 2012). Previous studies indicated that heterogeneity typically has nonlinear effects on biodiversity and ecosystem function, as moderate heterogeneity can maximise spatial asynchrony (Redon et al., 2014; Wilcox et al., 2017). However, our study did not observe nonlinear patterns between fragmentation per se and plant diversity and above-ground productivity. This may be due to the low spatial heterogeneity of this area as a result of agricultural intensification (Benton et al., 2003; Chen et al., 2019). The gradient of fragmentation per se in our study may not cover the optimal heterogeneity levels for maximising plant diversity and above-ground productivity (Thompson and Gonzalez, 2016).”

Meanwhile, we also discussed the nonlinear pattern of the BEF relationship with increasing levels of fragmentation per se to add depth to the discussion.

Line 442-451:

“In addition, our study found that the BEF relationship showed a nonlinear pattern with increasing levels of fragmentation per se. For a given level of habitat loss, the positive BEF relationship was strongest at moderate fragmentation per se level and became neutral at high fragmentation per se level. This can be explained by the increased spatial asynchrony at moderate fragmentation per se level, which can promote niche complementary among species in the community and thus strengthen the BEF relationship (Gonzalez et al., 2020; Thompson and Gonzalez, 2016; Tscharntke et al., 2012). The neutral BEF relationship at high fragmentation per se level may be due to edge effects enhancing environmental filtering, thereby leading to functional redundancy among species and decoupling the BEF relationship (Fetzer et al., 2015; Hu et al., 2016; Zambrano et al., 2019).”

(a) Line 74-75: I was wondering if you also thought of spatial insurance effects or spatial asynchrony effects that can emerge with habitat fragmentation, which could lead to increased ecosystem functioning as well? (refs. above).

Response: Thanks for your constructive suggestions. In the revised manuscript, we have explicitly considered the spatial insurance effect or spatial asynchrony as the important mechanism for fragmentation per se to increase plant diversity, ecosystem function, and the BEF relationship.

Line 74-77:

“In theory, habitat loss and fragmentation per se can regulate ecosystem function and the BEF relationship by altering species composition, interactions, and spatial asynchrony regardless of changes in species richness (Liu et al., 2018; Thompson and Gonzalez, 2016; Tscharntke et al., 2012).”

Line 402-408:

“In our study, a possible mechanism for the positive impacts of fragmentation per se on plant diversity and above-ground productivity (indirect positive impact via plant diversity) is that fragmentation per se increases the habitat heterogeneity in the landscape, which can promote biodiversity through spatial asynchrony and spatial insurance effects (Tscharntke et al., 2012). Previous studies indicated that heterogeneity typically has nonlinear effects on biodiversity and ecosystem function, as moderate heterogeneity can maximise spatial asynchrony (Redon et al., 2014; Wilcox et al., 2017).”

Line 442-451:

“In addition, our study found that the BEF relationship showed a nonlinear pattern with increasing levels of fragmentation per se. For a given level of habitat loss, the positive BEF relationship was strongest at moderate fragmentation per se level and became neutral at high fragmentation per se level. This can be explained by the increased spatial asynchrony at moderate fragmentation per se level, which can promote niche complementary among species in the community and thus strengthen the BEF relationship (Gonzalez et al., 2020; Thompson and Gonzalez, 2016; Tscharntke et al., 2012). The neutral BEF relationship at high fragmentation per se level may be due to edge effects enhancing environmental filtering, thereby leading to functional redundancy among species and decoupling the BEF relationship (Fetzer et al., 2015; Hu et al., 2016; Zambrano et al., 2019).”

(b) I was wondering, if this result of linear effects could also be the result of a fragmentation gradient that does not cover the whole range of potential values? Maybe it would be good to compare the gradient in habitat fragmentation in your study with a theoretical minimum maximum/considering that there might be an optimal medium degree of fragmentation.

Response: Thanks for your insightful comment. We agree with your point that the linear effect of fragmentation per se in our study may be due to the fact that the gradient of fragmentation per se in this region may not cover the optimal heterogeneity levels for maximising spatial asynchrony. This is mainly because the agricultural intensification in the agro-pastoral ecotone of northern China could lead to lower spatial heterogeneity in this region. We have explicitly discussed this point in the revised manuscript.

Line 406-413:

“Previous studies indicated that heterogeneity typically has nonlinear effects on biodiversity and ecosystem function, as moderate heterogeneity can maximise spatial asynchrony (Redon et al., 2014; Wilcox et al., 2017). However, our study did not observe nonlinear patterns between fragmentation per se and plant diversity and above-ground productivity. This may be due to the low spatial heterogeneity of this area as a result of agricultural intensification (Benton et al., 2003; Chen et al., 2019). The gradient of fragmentation per se in our study may not cover the optimal heterogeneity levels for maximising plant diversity and above-ground productivity (Thompson and Gonzalez, 2016).”

(4) Some additional suggestions:(a) Line 3: Maybe add "via reducing the percentage of grassland specialists in the community"?

Response: Thanks for your suggestion. We have revised this sentence.

Line 19:

“Habitat loss can weaken the positive BEF relationship via reducing the percentage of grassland specialists in the community”

(b) Lines 46-48: Maybe add "but see: Duffy, J.E., Godwin, C.M. & Cardinale, B.J. (2017). Biodiversity effects in the wild are common and as strong as key drivers of productivity. Nature."

Response: Thanks for your suggestion. We have added this reference here.

Line 47-49:

“When research expands from experiments to natural systems, however, BEF relationships remain unclear in the natural assembled communities, with significant context dependency (Hagan et al., 2021; van der Plas, 2019; but see Duffy et al., 2017).”

(c) Lines 82-87 and lines 90-93: Hence, your study actually is in contrast to these findings, i.e., fragmented landscapes do not necessarily have a lower fraction of grassland specialists? If yes, could you highlight this more explicitly?

Response: Thanks for your insightful comment. We have explicitly highlighted this point in the revised manuscript.

Line 434-439:

“Meanwhile, our study demonstrates that habitat loss, rather than fragmentation per se, can decrease the degree of habitat specialisation by leading to the replacement of specialists by generalists in the community, thus weakening the BEF relationship. This is mainly because fragmentation per se did not decrease the grassland specialist richness in this region, whereas habitat loss decreased the grassland specialist richness and led to the invasion of more weeds from the surrounding farmland into the grassland community (Yan et al., 2022; Yan et al., 2023).”

(d) Line 360: Could you add some examples of these multiple ecosystem functions you refer to?

Response: Thanks for your suggestion. We have added some examples of these multiple ecosystem functions here.

Line 456-457:

“Therefore, future studies are needed to focus on multiple ecosystem functions, such as below-ground productivity, litter decomposition, soil carbon stocks, etc.”

**Reviewer #3 (Public Review):**
Summary:The authors aim to solve how landscape context impacts the community BEF relationship. They found habitat loss and fragmentation per se have inconsistent effects on biodiversity and ecosystem function. Habitat loss rather than fragmentation per se can weaken the positive BEF relationship by decreasing the degree of habitat specialization of the community.Strengths:The authors provide a good background, and they have a good grasp of habitat fragmentation and BEF literature. A major strength of this study is separating the impacts of habitat loss and fragmentation per se using the convincing design selection of landscapes with different combinations of habitat amount and fragmentation per se. Another strength is considering the role of specialists and generalists in shaping the BEF relationship.

Response: We are grateful to you for the recognition and constructive comments. All the comments and suggestions are very constructive for improving this manuscript. We have carefully revised the manuscript following your suggestions. All changes are marked in red font in the revised manuscript.

Weaknesses:(1) The authors used five fragmentation metrics in their study. However, the choice of these fragmentation metrics was not well justified. The ecological significance of each fragmentation metric needs to be differentiated clearly. Also, these fragmentation metrics may be highly correlated with each other and redundant. I suggest author test the collinearity of these fragmentation metrics for influencing biodiversity and ecosystem function.

Response: Thanks for your constructive suggestion. The fragmentation metrics used in our study represent the different processes of breaking apart of habitat in the landscape, which are widely used by previous studies (Fahrig, 2003; Fahrig, 2017). In the revised manuscript, we have provided more detailed information about the ecological significance of these fragmentation indices.

Line 142-148:

“The patch density metric reflects the breaking apart of habitat in the landscape, which is a direct reflection of the definition of fragmentation per se (Fahrig et al., 2019). The edge density metric reflects the magnitude of the edge effect caused by fragmentation (Fahrig, 2017). The mean patch area metric and the mean nearest-neighbor distance metric are associated with the area and distance effects of island biogeography, respectively, reflecting the processes of local extinction and dispersal of species in the landscape (Fletcher et al., 2018).”

Meanwhile, we have calculated the variance inflation factors (VIF) for each fragmentation metric to assess their collinearity. The VIF of these fragmentation metrics were all less than four, suggesting no significant multicollinearity for influencing biodiversity and ecosystem function.

**Author response table 2. sa4table2:** Variance inflation factors of habitat loss and fragmentation per se indices for influencing plant richness and above-ground biomass.

Variable	Variance inflation factors (VIF)
Patch density	2.28
Edge density	1.21
Mean patch area	2.54
Mean nearest-neighbor distance	1.25

(2) I found the local environmental factors were not considered in the study. As the author mentioned in the manuscript, temperature and water also have important impacts on biodiversity and ecosystem function in the natural ecosystem. I suggest authors include the environmental factors in the data analysis to control their potential impact, especially the structural equation model.

Response: Thanks for your constructive suggestion. We agree with you that environmental factors should be considered in our study. In the revised manuscript, we have integrated two environmental factors related to water and temperature (soil water content and land surface temperature) into the data analysis to control their potential impact. The main results and conclusions of the revised manuscript are consistent with those of the previous manuscript.

**Reviewer #3 (Recommendations For The Authors):**
(1) L60-63. The necessity to distinguish between habitat loss and fragmentation per se is not clearly stated. More information about biodiversity conservation strategies can be given here.

Response: Thanks for your suggestion. In the revised manuscript, we have provided more evidence about the importance of distinguishing between habitat loss and fragmentation per se for biodiversity conservation.

Line 62-67:

“Habitat loss is often considered the major near-term threat to the biodiversity of terrestrial ecosystems (Chase et al., 2020; Haddad et al., 2015), while the impact of fragmentation per se remains debated (Fletcher Jr et al., 2023; Miller-Rushing et al., 2019). Thus, habitat loss and fragmentation per se may have inconsistent ecological consequences and should be considered simultaneously to establish effective conservation strategies in fragmented landscapes (Fahrig et al., 2019; Fletcher et al., 2018; Miller-Rushing et al., 2019).”

(2) L73-77. The two sentences are hard to follow. Please rephrase to improve the logic. And I don't understand the "however" here. There is no twist.

Response: Thanks for your suggestion. We have rephrased the two sentences to improve their logic.

Line 74-79:

“In theory, habitat loss and fragmentation per se can regulate ecosystem function and the BEF relationship by altering species composition, interactions, and spatial asynchrony regardless of changes in species richness (Liu et al., 2018; Thompson and Gonzalez, 2016; Tscharntke et al., 2012). This is because species in communities are not ecologically equivalent and may respond differently to habitat loss and fragmentation per se, and contribute unequally to ecosystem function (Devictor et al., 2008; Wardle and Zackrisson, 2005).”

(3) L97. Are grasslands really the largest terrestrial ecosystem? Isn't it the forest?

Response: We apologize for making this confusion. We have rephrased this sentence here.

Line 101-104:

“Grasslands have received considerably less attention, despite being one of the largest terrestrial ecosystems, and suffering severe fragmentation due to human activities, such as agricultural reclamation and urbanisation (Fardila et al., 2017).”

(4) Fig.1, whether the four sample plots presented in panel b are from panel a. Please add the scale bar in panel b.

Response: Thanks for your comment. The four sample plots presented in panel b are from panel a in Figure 1. We have also added the scale bar in panel b.

(5) L105. This statement is too specific. Please remove and consider merging this paragraph with the next.

Response: Thanks for your suggestion. We have removed this sentence and merged this paragraph with the next.

(6) L157. The accuracy and kappa value of the supervised classification should be given.

Response: Thanks for your suggestion. We have added the accuracy and kappa value of the supervised classification in the revised manuscript.

Line 176-177:

“The overall classification accuracy was 84.3 %, and the kappa coefficient was 0.81.”

(7) I would recommend the authors provide the list of generalists and specialists surveyed in the supplementary. Readers may not be familiar with the plant species composition in this area.

Response: Thanks for your suggestion. We agree with your point. We have provided the list of generalists and specialists surveyed in the Appendix Table A4.

Line 282-283:

“A total of 130 vascular plant species were identified in our study sites, including 91 grassland specialists and 39 weeds (Appendix Table A4).”

(8) Fig.4, it is better to add the results of variation partition to present the relative contribution of habitat fragmentation, environmental factors, and plant diversity.

Response: Thanks for your suggestion. We have integrated the landscape context, environmental factors, and plant diversity into the multi-model averaging analysis and redraw Figure 4 to present their relative importance for above-ground biomass.

Line 313-319:

**Author response image 3. sa4fig3:** Standardised parameter estimates and 95% confidence intervals for landscape context, plant diversity, and environmental factors affecting above-ground biomass from 130 landscapes in the Tabu River Basin, a typical agro-pastoral ecotone of northern China. Standardised estimates and 95% confidence intervals are calculated by the multi-model averaging method based on the four optimal models affecting above-ground biomass (Appendix Table A3). ** represent significance at the 0.01 level.

(9) Please redraw Fig.2 and Fig.5 to integrate the environmental factors. Add the R-square to Fig 5.

Response: Thanks for your suggestion. We have integrated two environmental factors into the structural equation model and redraw Figure 2 and Figure 5 in the revised manuscript. And we have added the R-square to the Figure 5.

(10) L354. The authors should be careful to claim that habitat loss could reduce the importance of plant diversity to ecosystem function. This pattern observed may depend on the type of ecosystem function studied.

Response: Thanks for your suggestion. We have avoided this claim in the revised manuscript and explicitly discussed the importance of simultaneously focusing on multiple ecosystem functions, such as below-ground productivity, litter decomposition, soil carbon stocks, etc.

Line 454-457:

“This inconsistency can be explained by trade-offs between different ecosystem functions that may differ in their response to fragmentation per se (Banks-Leite et al., 2020). Therefore, future studies are needed to focus on multiple ecosystem functions, such as below-ground productivity, litter decomposition, soil carbon stocks, etc.”